# Developing Chatbots for Cyber Security: Assessing Threats through Sentiment Analysis on Social Media

**Amit Arora [1,*], Anshu Arora [1] and John McIntyre [2]**

[1] School of Business and Public Administration, University of the District of Columbia, Washington, DC 20008, USA; anshu.arora@udc.edu

[2] Scheller College of Business, Georgia Institute of Technology, Atlanta, GA 30332, USA; john.mcintyre@scheller.gatech.edu

\* Correspondence: amit.arora@udc.edu

**Abstract:** In recent years, groups of cyber criminals/hackers have carried out cyber-attacks using various tactics with the goal of destabilizing web services in a specific context for which they are motivated. Predicting these attacks is a critical task that assists in determining what actions should be taken to mitigate the effects of such attacks and to prevent them in the future. Although there are programs to detect security concerns on the internet, there is currently no system that can anticipate or foretell whether the attacks will be successful. This research aims to develop sustainable strategies to reduce threats, vulnerability, and data manipulation of chatbots, consequently improving cyber security. To achieve this goal, we develop a conversational chatbot, an application that uses artificial intelligence (AI) to communicate, and deploy it on social media sites (e.g., Twitter) for cyber security purposes. Chatbots have the capacity to consume large amounts of information and give an appropriate response in an efficient and timely manner, thus rendering them useful in predicting threats emanating from social media. The research utilizes sentiment analysis strategy by employing chatbots on Twitter (and analyzing Twitter data) for predicting future threats and cyber-attacks. The strategy is based on a daily collection of tweets from two types of users: those who use the platform to voice their opinions on important and relevant subjects, and those who use it to share information on cyber security attacks. The research provides tools and strategies for developing chatbots that can be used for assessing cyber threats on social media through sentiment analysis leading to a global sustainable development of businesses. Future research may utilize and improvise on the tools and strategies suggested in our research to strengthen the knowledge domain of chatbots, cyber security, and social media.

**Keywords:** chatbot; cyber security; artificial intelligence; threats; vulnerability; data manipulation; social media

## 1. Introduction

Cybersecurity is a multidisciplinary field and can have far-reaching economic, environmental, and social consequences [1–3]. Cybersecurity statistics indicate that there are 2200 cyber-attacks per day, with a cyber-attack happening every 39 s on average. In the US, a single data breach costs an average of USD 9.44 million, and cybercrime is predicted to cost USD 8 trillion in 2023 [4]. As governments and businesses become more reliant on new communication technologies and social media, the threat of cyber-attacks on such organizations has increased tremendously. To counter such threats, governments and businesses have increased their investments in cybersecurity [5]. Advances in natural language processing (NLP) and machine learning (ML) techniques have led to chatbots (also known as conversational agents) becoming capable of extracting meaningful information regarding cybersecurity threats [6] on social media. The rapid deployment of artificial intelligence (AI) coupled with the digitalization of a globalized economy has produced a vast amount of textual data through social media. Chatbot applications along with technology-enabled

solutions lead to the sustainable development of global businesses and economies. Governments, businesses, and political parties depend on the sentiments and opinions expressed on social media sites to gauge the mood of the public in real time [7]. This is also a vital source of information related to security threats to a nation and to business organizations. Consequently, it becomes imperative for intelligence and security communities to delve deeper into cybersecurity to protect national security and economic interests.

Social networks on the internet have enabled people to interact with each other in real-time. Microblogging platforms, such as Twitter, have emerged as the most popular communication tool since it allows a wide variety of expressions, such as interactive short texts, pictures, emojis, etc., with relative ease [8,9]. Such platforms act as a social public square where users express their feelings, sentiments, ideas, and opinions on wide-ranging topics. Research has shown that analyzing these feelings and sentiments expressed on social networks and platforms is an effective way to forecast a variety of events such as market trends, election results, brand image, etc. [8,10]. Sentiment analysis can be performed quickly on a large amount of textual data available on social platforms and has been applied in various fields. Recent research has focused on the sentiment analysis of text data on social media related to COVID-19 and monkeypox [11], as well as business-related entrepreneurship [12]. However, there is a dearth of research to assess sentiments in detecting probable cybersecurity threats.

Previous research has shown that the best practice to combat threats in cyber security is to develop strategies that are complementary to each specific threat [13]. In this research, we develop strategies to reduce the threats, vulnerabilities, and data manipulation of chatbots, consequently improving cyber security. Specifically, we develop a chatbot on Bot Libre, an open-source platform, and deploy it on Twitter. The research also focuses on sentiment analysis of the tweets generated by Twitter users conversing with our developed chatbot.

The objectives of this research are as follows: (1) survey the existing state-of-the-art multilingual chatbot tools, (2) develop and test this Chatbot Testbed on Twitter, (3) conduct sentiment analysis of textual data generated through tweets, and (4) create documentation and materials so that this toolbox can be used by a variety of users with sustainable development goals. In pursuance of these objectives, we investigate the following research questions (RQs):

RQ1. What are the different types of chatbots and chatbot tools available to counter or neutralize cybersecurity threats that target human vulnerabilities?
RQ2. How can chatbots be developed and tested for cybersecurity using social media?
RQ3. How can one assess these chatbots and their effectiveness for cybersecurity on Twitter?

RQ1 is linked to Research Objective 1; RQ2 is linked to Research Objectives 1 and 2; and RQ3 is linked to Research Objectives 3 and 4. The Chatbot Testbed was created by integrating existing open-source and commercial tools to effectively create a solution that is usable for understanding problems in influence, information operations, and insider threat. In this research, we utilize the theory of planned behavior [14–16] as the classical theoretical framework for various types of chatbots, the role of chatbots in training, uses of chatbots, chatbot framework, and implementation of chatbots. Organizations implementing social media chatbots for cybersecurity reap the benefits when organizational employees respect and adhere to information security, according to the theory of planned behavior [17].

## 2. Background and Related Work

A chatbot is an application that uses artificial intelligence (AI) to communicate. Artificial intelligence is the automation of intelligent behavior which allows machines to simulate anthropomorphic conversations. Chatbots have been programmed to use artificial intelligence and concepts such as natural language processing (NLP), artificial intelligence markup language (AIML), pattern matching, chat script, and natural language understanding (NLU) to communicate with users, analyze the conversation, and use the extracted data for marketing, personal content, to target specific groups, etc. The knowledge domain, the service provided, the goals, the input processing and response generation method, the

human-aid, and the build method are some of the categories under which chatbots can be classified.

The knowledge domain classification considers the knowledge a chatbot can access, as well as the amount of data it is trained on. Closed-domain chatbots are focused on a certain knowledge subject and may fail to answer other questions, but open-domain chatbots can talk about various topics and respond effectively [18]. Conversely, the sentimental proximity of the chatbot to the user, the quantity of intimate connection, and chatbot performance are factors in the classification of chatbots based on the service provided. Interpersonal chatbots are in the communication area and offer services such as restaurant reservations, flight reservations, and FAQs. They gather information and pass it on to the user, but they are not the user's companions. They are permitted to have a personality, be nice, and recall information about the user; however, they are not required or expected to do so [18]. Adamopoulou et al. [18] states that "Intrapersonal chatbots exist within the personal domain of the user, such as chat apps like Messenger, Slack, and WhatsApp. They are companions to the user and understand the user like a human does. Inter-agent chatbots become omnipresent while all chatbots will require some inter-chatbot communication possibilities. The need for protocols for inter-chatbot communication has already emerged. Alexa-Cortana integration is an example of inter-agent communication" (pp. 373–383).

Informative chatbots, such as FAQ chatbots, are designed to offer the user information that has been stored in advance or is available from a fixed source. The manner of processing inputs and creating responses is taken into consideration when classifying based on input processing and response generation. The relevant replies are generated using one of three models: rule-based, retrieval-based, and generative. Another classification for chatbots is based on how much human-aid is included in its components. Human computation is used in at least one element of a human-aid chatbot. To address the gaps produced by the constraints of completely automated chatbots, crowd workers, freelancers, or full-time employees can incorporate their intelligence in the chatbot logic. The work in [18] (pp. 373–383) examines the main classification of chatbots as per the development platform permissions, where the authors defined 'development platforms' as "...open-source, such as RASA, or can be of proprietary code such as development platforms typically offered by large companies such as Google or IBM."

Two of the main categories that chatbots may fall into as it relates to their anthropomorphic characteristics are the error-free and the clarification chatbot. Anthropomorphism is "the attribution of human characteristics or traits to nonhuman agents" [19] (p. 865). Anthropomorphic chatbots are perceived to be more palatable to consumers since consumers perceive the chatbots to be humanlike, rather than how firms design chatbots as humanlike [20]. An error-free chatbot can be defined as a hypothetically flawless chatbot, while a clarification chatbot has difficulties inferring meaning and therefore asks for clarification from the user. Clarification chatbots are seen as more anthropomorphic since clarification by the chatbot is seen as giving care and attention to the needs of the customer. According to [21], "The error-free chatbot offers no indication that it is anything but human. It correctly interprets all human utterances and responds with relevant and precise humanlike utterances of its own." On the first parse, the clarification chatbot does not have the intelligence to accurately interpret all human utterances. The chatbot, on the other hand, is clever enough to identify the root of the misunderstanding, referred to as a difficulty source, and request an explanation. Since seeking clarification is a normal element of interpersonal communication, clarification chatbots' anthropomorphic characteristics increase with their ability to recognize a problem source and display intersubjective effort. There is no current commercial application of the error-free chatbot; however, clarification chatbots are currently being used by companies such as Amazon, Walmart, T-Mobile, Bank of America, and Apple, as first-contact customer service representatives.

Threats and vulnerability are key factors (and dangers) affecting the cyber security of chatbots. Cyber threats can be characterized as methods in which a computer system can be hacked. Spoofing, tampering, repudiation, information disclosure, denial of service, privi-

lege elevation, and other threats are examples of chatbot threats. Conversely, vulnerabilities are ways in which a system can be harmed that are not appropriately mitigated. When a system is not effectively maintained, has bad coding, lacks protection, or is subject to human mistakes, it becomes vulnerable and accessible to assaults. Self-destructive messages can be used in conjunction with other security measures such as end-to-end encryption, secure protocol, user identity authentication, and authorization to reduce vulnerabilities. Another method to ensure the security of chatbots is the use of user behavioral analytics (UBA).

A vulnerability is defined as a weakness in a system's security protocols, internal controls, or implementation that could be exploited or activated by a threat source. The secure development lifecycle refers to the process of incorporating security components into the software development lifecycle (SDLC). SDLC, on the other hand, is a thorough plan that outlines how companies construct applications from conception through decommission. According to [13], implementing security development lifecycle (SDL)-related activities into the development lifecycle is one of the most effective ways to mitigate vulnerabilities. Planning and needs, testing the code and outcomes, architecture and design, test planning, and coding are phases commonly followed by all models for the secure development lifecycle. This reduces the vulnerabilities and openness to attacks. User behavioral analytics (UBA) is a method of analyzing user activity patterns through the use of software applications. It allows for the use of advanced algorithms and statistical analysis to spot any unusual behavior that could be a security risk. The use of this analytical software will allow for easy identification of other bots being used to infiltrate a secure system through hacking. Hence, this reduces the risk of a cyber-attack.

As previously mentioned, cyber threats can be characterized as methods in which a computer system can be hacked. Spoofing, tampering, repudiation, information disclosure, denial of service, and privilege elevation are examples of threats. To reduce the impacts of these threats, specific approaches need to be taken for each particular threat. Spoofing is performed to gain information and use it for the impersonation of something or someone else. To abate this, correct authentication such as a strong password is required to secure sensitive data. Tampering is a threat where the hacker aims to maliciously modify data. Here, the mitigation strategy is to use digital signatures, audit trails, a network time protocol, and log timestamps. Denial of service is another category of threats in which the attacker intends to deny access to valid users. In this instance, the best strategies to reduce this threat are filtering and throttling [13].

As of December 2022, Twitter had 368 million monthly active users worldwide (statista.com/statistics/303681/twitter-users-worldwide (accessed on 16 July 2023)), providing a chance to gather a large amount of data in near-real time. In this research, we focus on the development and deployment of a chatbot on a social media platform in order to collect a large sample of textual data in the form of tweets and perform sentiment analysis using algorithmic techniques to forecast certain threats and vulnerabilities related to cybersecurity.

Sentiment analysis of user-generated data (e.g., tweets generated by users) is becoming increasingly popular as a research focus in multi-disciplinary fields. Previous research has focused on sentiment analysis of Twitter data across a broad range of topics ranging from emotions expressed by users [22], opinion mining on cryptocurrency [23], exploring challenges of remote work [24], customer satisfaction in the airline industry [25], etc. Our research strives to make several contributions to theory and practice. To the best of our knowledge, previous research has not focused on providing a flexible tool for the development and deployment of a chatbot on Twitter that can converse with other users and use this information to detect possible cybersecurity threats emanating from sentiments expressed by such users. Our research focuses on the combination of both of these aspects in a single research work. Furthermore, we utilize the theory of planned behavior as the classical theoretical framework to position our research in addition to using the Design of Experiments as the research design methodology.

## 3. Methods and Data Analysis

This section focuses on the two main aspects of the research: a) development and deployment of a conversational chatbot on a social media site; and b) conducting sentiment analysis on the vast amount of textual data from a social media site.

We used the Design of Experiments as the research design methodology to focus on the above two aspects of the research. There are three main components in the Design of Experiments—input, output, and process which transforms inputs into outputs [26]. A simple example could be in the context of a manufacturing process where inputs are factors such as raw material, machines, standard operating procedures, manpower, etc. Outputs are a particular product/service with specified quality/characteristics. In the context of our research, the input is the raw text data obtained from Twitter, the process is the sentiment analysis, and the output is the sentiment classification as positive, neutral, or negative. The output in this research depends on (a) controllable factors such as algorithm selection and parameters; and (b) uncontrollable factors such as data noise and the randomness of data sampling [27].

### 3.1. Development and Deployment of the Chatbot

Initially, the project team focused on building a chatbot on the SAP open-source platform. However, it is hard to use the SAP conversational AI chatbot outside of the SAP S/4 Hana cloud. After considering other open-source platforms like Botpress, our conversational chatbot was developed on Bot Libre, an open-source end-to-end chatbot-building platform. It can be used to build, train, connect, and monitor the chatbot on a social media site. The Bot Libre chatbot uses both text and images and is categorized as a communication channels chatbot [13,18]. This platform allows the chatbot to be deployed on various social media sites like Twitter, WhatsApp, Facebook, Discord, Kik, etc. The language modeling, which is a part of personalizing how the bot communicates with specific users, allows the bot to interact with users in multiple languages and can be tailored to include English, French, Russian Spanish, Italian, and Japanese, among other languages.

Currently, our chatbot can only converse in the English language on Twitter. The automation feature allows the bot to tweet over an extensive period. For example, in the month of March, the chatbot was programmed to tweet "Happy Women's History Month" every 24 h. It utilizes the 'conversational feature' by initiating and maintaining conversations with other users of Twitter. Its 'informational effect' and 'data effect' are highlighted by its ability to collect data from conversations it has with other users, as well as to extract information from the platform based on key terms searched. For example, the chatbot can search the key terms "Putin", "nuclear weapon", and "Russia" and extract all tweets associated with these key terms. The goal of the chatbot is to communicate and extract information/intelligence from users on Twitter which can be used by intelligence and security communities. Any keyword that can be deemed as a threat (e.g., hate speech, defense-related, etc.) can be searched on the Twitter platform using the chatbot. The information is collected using the application programming interface (API) keys. This monitoring of information on a social media platform will aid in cyber security within the United States. The analytics feature of the Bot Libre platform can provide useful information about the chat conversations conducted by the chatbot during a specific day, week, month, or any specified period. Figure 1 illustrates the analytics feature of the Bot Libre platform. Data that can be analyzed include conversations, messages, conversation length, response time, connects, chats, errors, etc.

Next, the project team focused on conducting sentiment analysis on the vast amount of textual data collected from Twitter.

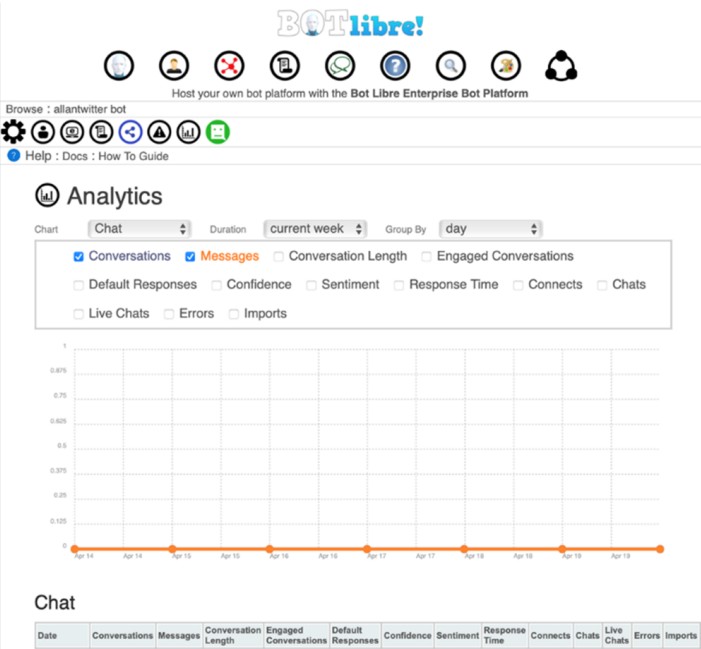

**Figure 1.** Snapshot of analytics feature of the chatbot developed on the Bot Libre platform.

### 3.2. Sentiment Analysis

Previous research has shown that written text on social media sites is impacted by the emotions, intentions, and thoughts of the user [28,29]. Thus, written text is a useful source of information about the user. This section describes the process of data collection, cleaning, and analysis in detail.

#### 3.2.1. Data Collection

First, we discuss the collection of data. The Twitter API academic research access was used to collect global, real-time, and historical textual data in the form of tweets. The collected tweets were processed in JSON and added to corpus *C* [8,30], identified as:

$$\mathcal{C} = c_i \in \left\{ tweet_{id}, tweet_{text_i}, tweet_{date_i}, tweet_{language_i} \right\}_{i=1}^{n} \forall t_{i,4} = en \qquad (1)$$

$c_i$ represents the $i^{th}$ tweet in the corpus. Each tweet is identified by four objects: id, text, date, and language. *C* is stored locally in MySQL, a relational database. A primary key, $tweet_{id}$, is allocated to each tweet in *C*, which is utilized to identify unprocessed tweets.

The code for Twitter API credentials and extraction of tweets is given in the Appendix A.

#### 3.2.2. Data Pre-Processing

The next step is data pre-processing, which consists of speech tagging and noise removal. Since the tweets posted by users on Twitter are informal in nature, the raw textual data in the form of tweets tends to have grammatical errors, may be unstructured, and can generally be considered noisy. This could potentially make it difficult to interpret the data correctly. Hence, the data should be pre-processed before analyzing and determining the user's sentiment.

(a)　Speech tagging is a process of assigning a specific tag to each word in the corpus of tweets. These tags divide the textual data, e.g., tweets, into verbs, adverbs, nouns, adjectives, etc., which can be used as potential markers to determine the polarity (or sentiment) of tweets. The polarity (or sentiment) can be positive, neutral, or negative. Exclamation marks, question marks, and emoticons are also considered for determining polarity. The tags utilized in this study are an optimized version of Penn

Treebank's compendium [31]. Below is an example of the speech tagging of a sample tweet [8] (Figure 2).

I hate #ISIS and everything that they stand for .
O   V    ^    &    N        P   O    V    P   ,

| Sample | Tag | Attribute |
|--------|-----|-----------|
| I | O | pronoun (personal / no possessive) |
| hate | V | copulative verb |
| #ISIS | ^ | proper name |
| and | & | coordinating conjunction |
| everything | N | common noun |
| that | P | preposition |
| they | O | particle verb |
| stand | V | copulative verb |
| for | P | preposition |
| , | , | punctuation |

**Figure 2.** Example of speech tagging of a tweet.

(b) Next, we focus on the removal of noise from data. Sometimes, tweets may include text and other markers that seem to be unrelated to the expressed sentiment, which need to be removed before data analysis. Noise in twitter data may be in the form of URLs, replies to other users, retweets, and common stop words which do not add value to the meaning of a tweet [8,32]. To eliminate these occurrences, a noise removal procedure is used which includes the removal of @mentions, symbols such as '#' and ':', and hyperlinks. Pre-processing of the corpus of tweets also included removing retweets (or reposts of original tweets). Each tweet in the corpus has a unique identification number which is retained each time the original one is retweeted.

(c) Lemmatization is the process of obtaining lemmas of words from the corpus of tweets. A lemma is a word as it is presented in a dictionary. For example, the lemma for the words 'run', 'ran', 'running', 'runs' is run. In TextBlob, the lemmatization process is based on WordNet developed by Princeton University [33] which is an open-source database.

The code used for data pre-processing is given in Appendix A.

### 3.2.3. Sentiment Extraction

The process of determining the emotional tone (positive, negative, or neutral) in a body of text is commonly referred to as sentiment extraction. This is performed by locating and identifying candidate markers written by most users in their tweets. To achieve this, the Apriori algorithm is utilized. This is an algorithm for mining frequent item sets and learning association rules [8,34]. If a collection of textual data contains a minimum of 1% support as the frequency of occurrence, it is classified as featuring frequent markers [8,9]. The goal of association rules is to uncover latent words that can be used as frequent markers.

$$\Psi^f = \{\psi : \psi \in \Psi \wedge minSupport(\psi)\} \tag{2}$$

According to the methods presented in [8,35], candidate markers that the Apriori algorithm cannot identify are removed.

The National Initiative for Cybersecurity Careers and Studies (NICCS) lexicon dictionary [36] δ for information security is used as a reference to identify tweets related to cybersecurity. This dictionary contains key cybersecurity terms which provide a comprehensive understanding of the definitions/terminology pertaining to cybersecurity [8,37].

The following is computed [8,10] to acquire the samples that contain words in δ:

$$\mathcal{H}_i = \left\{ \Psi_{i,1}, \Psi_{j,2} \right\} \text{ if } \delta_i\left(\Psi_{j,2}\right) > 1 \tag{3}$$

Or the following:

$$\mathcal{B}_i = \left\{ \Psi_{i,1}, \Psi_{j,2} \right\} \tag{4}$$

where $\mathcal{H}$ is the collection of samples that contain at least one word from δ and $\mathcal{B}$ is the collection of the remaining samples, i.e., tweets that do not contain specific content about security issues:

$$\mathcal{H} = \{\psi : \psi \in \Psi \wedge \delta(\psi) > 1\} \tag{5}$$

$$\mathcal{B} = \{\psi : \psi \in \Psi \wedge \delta(\psi) < 1\} \tag{6}$$

3.2.4. Sentiment Orientation and Analysis

The sentiment orientation stage is carried out by analyzing the frequent markers in $\Psi_f$ in $\mathcal{H}_i$ and $\mathcal{B}_i$ where $i$ is the $i'^{th}$ sample. The polarity is determined by scores previously defined in the SentiWordnet compendium [8,38]. SentiWordnet is a lexical resource compendium for opinion mining. Each word set consists of three sentiments: positive, negative, and neutral [8,39]. These sentiments are based on relationships and associations between words such as antonyms, synonyms, and hyponyms. These are used to develop certain rules to identify the polarity (or sentiment) of the text in consideration [8,40].

The findOrientation algorithm is used to identify tweets having a negative orientation, with $\mathcal{H}$ and $\mathcal{B}$ being the only compendiums containing negative markers.

The algorithm is given in Appendix A.

The $\mathcal{H}^1$ and $\mathcal{B}^1$ compendiums obtained using the findOrientation algorithm are linked to their respective primary key $tweet_{idi}$ of corpus $C$, which is utilized to establish the original tweet's creation date, $tweet_{datei}$.

The tweets are sorted by date and then combined to obtain a daily total score for $\mathcal{H}^1$ and $\mathcal{B}^1$ as shown below [8,41]:

$$\sum_{p=1}^{n} \mathcal{H}', \sum_{p=1}^{n} \mathcal{B}' \tag{7}$$
$$n = \text{number of tweets per day}$$

Sentiment analysis was performed on a sample of Twitter text. Google Colaboratory was used as our platform for machine-learning specific code in the Python language. The consumer key, consumer secret, access token, access token secret, and bearer token were downloaded from the Twitter project account with academic access and stored in a .csv file. These are necessary to give permission to retrieve the tweets needed for our analysis. The Tweepy Python library was imported for reducing the amount of code that it takes to perform certain actions, such as authentication, to allow access to the Tweets from the internal Twitter database.

For conducting sentiment analysis of the retrieved twitter texts, we used the lexicon-based approaches with open-source models/libraries. These include TextBlob and Valence Aware Dictionary and Sentiment Reasoner (VADER). VADER is a natural language toolkit package (NLTK) and calculates only the polarity (negative/positive) and the intensity (strength) of the emotion in a text. VADER assigns a score to each word in the text and then computes the compound score with a value ranging from −1 to +1 by adding and normalizing each valence score. In the output scores, −1 represents very negative, +1 is very positive, and 0 represents neutral. However, TextBlob outperforms VADER [25] and calculates the subjectivity of a sentence in addition to calculating its polarity. Subjectivity ranges from 0 (personal opinion) to 1 (factual information). Polarity ranges from −1 to +1, the same as in VADER [42].

We used TextBlob (https://textblob.readthedocs.io/en/dev/ (accessed on 19 May 2023)), a Python library, for processing and analyzing the Twitter data. It provides an open-source

API for speech tagging, sentiment analysis, etc. A text message or a tweet is a collection of words. Each word has its own intensity and semantic orientation that defines its sentiment. The overall sentiment of a text is calculated by taking the average of the sentiments of all words in the text. Sentiment is a function of polarity and subjectivity. The polarity of a text is a floating value between $[-1, 1]$ where $-1$ represents a highly negative sentiment and 1 represents a highly positive sentiment. A 0 value indicates a neutral sentiment. The subjectivity of a text is a floating value between $[0, 1]$ where 0 represents the least subjective and highly factual text, while 1 represents the most subjective and least factual text. Subjectivity is quantified as a measure of the amount of personal opinion vs. factual information in a text. TextBlob supports this complex analysis on text data and returns both polarity and subjectivity of a text.

The code used to classify subjectivity and polarity, and to visualize the words of a tweet in the form of a word cloud is given in Appendix A. A word cloud is based on the frequency of words in a text (e.g., for a collection of tweets). Also included in Appendix A is the code of a scatter plot of the subjectivity vs. polarity of tweets.

Figure 3 illustrates an example of a word cloud created from the most prominent words from the Twitter text data. As seen in Figure 3, the word cloud was created during March 2022 when the Russia and Ukraine war had started, leading to global tensions worldwide. The Russian invasion of Ukraine, also referred to as the Russian–Ukrainian conflict, continues to be a massive humanitarian catastrophe globally, and especially for people residing in Ukraine. We captured the tweet data at that time and found that "Russia" and "Ukraine" were the prominent words in tweets. Other popular words in tweets were "Putin", "Zelensky", "NATO", and "invasion", highlighting global tensions.

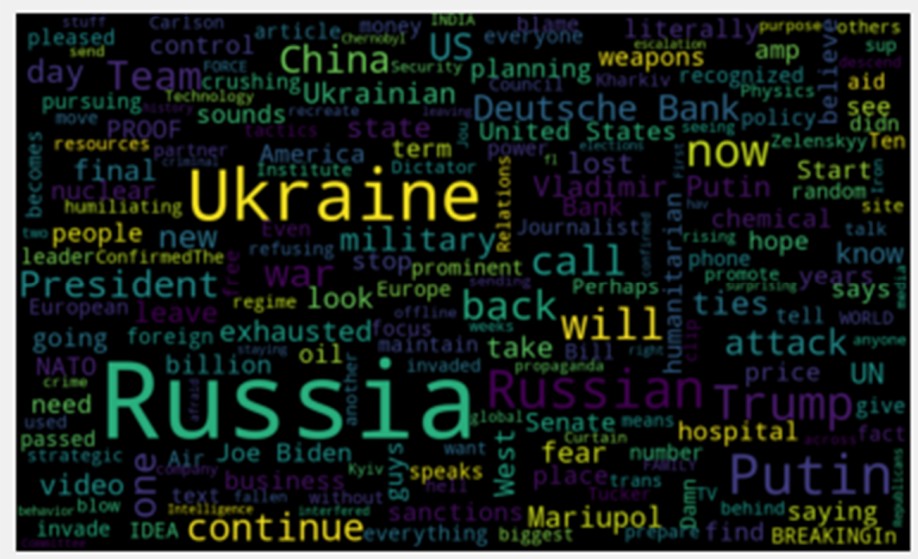

**Figure 3.** Word cloud created from prominent words in tweets.

Figures 4 and 5 illustrate an example of a scatter plot created with the subjectivity and polarity of a sample of tweets, and a bar graph representing the count of neutral, positive, and negative tweets. The area of interest from a cybersecurity point of view is the upper left quadrant in Figure 4 which represents specific tweets high on subjectivity and having a negative polarity. These may be referred to as outliers in the data which deserve a deeper analysis.

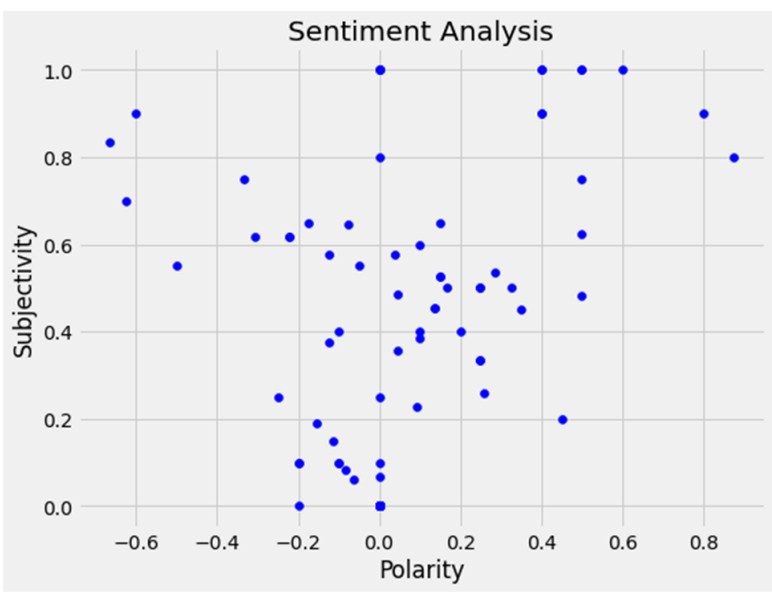

**Figure 4.** Subjectivity vs. polarity of a sample of tweets.

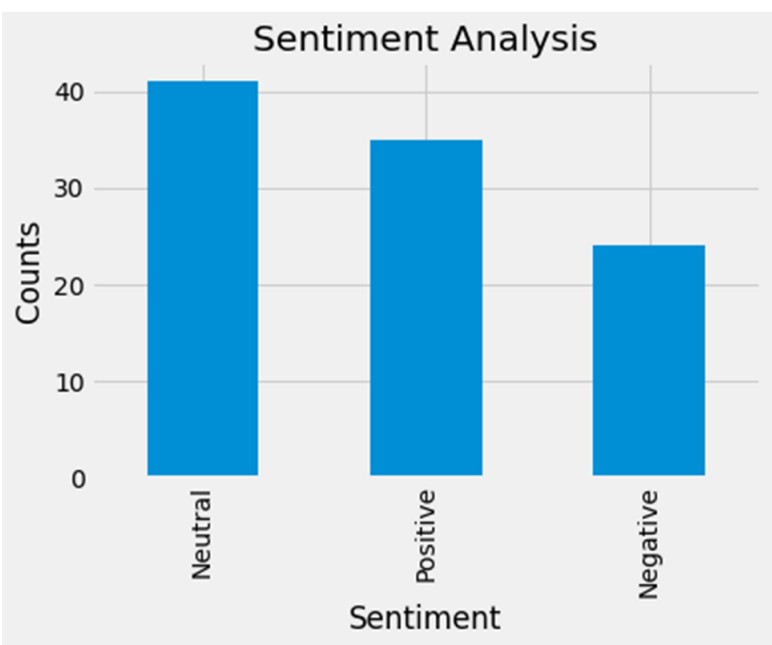

**Figure 5.** Classification of a sample of tweets.

In order to comprehend Figures 4 and 5, we provide examples of neutral, positive, and negative tweets during March 2022 (refer to Table 1 below). During March 2022, when tensions between Ukraine and Russia escalated, social media became a battleground for individuals to share their opinions and influence others. Twitter, in particular, was flooded with tweets related to the Russian–Ukraine war. Our findings revealed a significant increase in war-related tweets from the period of rising tension to the invasion and subsequent days. Additionally, sentiment analysis showed that there were more neutral tweets compared to positive and negative sentiment tweets.

**Table 1.** Examples of neutral, positive, and negative tweets during March 2022.

| | |
|---|---|
| **Neutral Sentiment** | I have enough problems of my own. I don't care about the war really |
| (no sentiments about war or its outcome) | War or No War: my problems are not going anywhere |
| | I don't care about the war outcome |
| **Positive Sentiment** | I hope that the Russian-Ukraine war ends amicably #Peace |
| (evidenced by peace, world order, stopping war) | Ukraine has put a brave front #StopWar #Peace |
| | Happy to see that the oppressed is not giving up to his aggressor! #IStandWithUkraine #NATO #WorldPeace |
| **Negative Sentiment** | #Terror #PutinWarCriminal #StopRussianAggression #StopRussia #StopPutin #Terrorist |
| (evidenced by fear, anger, and opposition to the Russian invasion of Ukraine) | I am Angry! I am Fearful about the outcome of the war #StopRussia |
| | #IOpposeRussia; #IOpposePutin; #PutinBeDa..ed |

## 4. General Discussion

This research addresses questions pertaining to different types of chatbots and chatbot tools available to counter or neutralize cybersecurity threats that target human vulnerabilities along with highlighting and assessing the effectiveness of chatbots (and chatbot development) for cybersecurity using Twitter data. In relation to the research questions, our research objectives include surveying the existing state-of-the-art multilingual chatbot tools, developing the Chatbot Testbed on Twitter, conducting sentiment analysis of textual data generated through tweets, and finally, creating the documentation so that this toolbox can be used by a variety of users with sustainable development goals. Using the theory of planned behavior, this research develops chatbots that can be used for assessing cyber threats on social media through sentiment analysis leading to a global sustainable development of businesses. Our Bot Libre chatbot, developed on an open-source platform, is deployed on Twitter and helps in the sentiment analysis of tweets generated by Twitter users conversing with our developed chatbot, thus assessing issues (risks and challenges) of cybersecurity and national security. Social media chatbots are helpful for global businesses for sending mass messages and updates. These conversational agents help to better understand consumer preferences (both online and offline), along with enhancing customer interaction rates through social media chatbots.

This study was conducted during March 2022 during the Russian invasion of Ukraine and highlighted the role of social media in modern-day warfare, where conflict occurs in both the physical and information environments. We employed sentiment analysis to understand how chatbot development and activity influences wider online discourse. For conducting sentiment analysis of the retrieved twitter texts, we used lexicon-based approaches with open-source models/libraries. We used TextBlob (https://textblob.readthedocs.io/en/dev/ (accessed on 19 May 2023)), a Python library, for processing and analyzing the Twitter data. It provides an open-source API for speech tagging, sentiment analysis, etc. Using the Design of Experiments research methodology, the study found that bot activity drives an increase in conversations surrounding fear, angst, positivity regarding outcomes of war/conflict, neutral emotions, and the emotions surrounding work/governance. Our findings revealed a significant increase in war-related tweets from the period of rising tension to the invasion. We employed sentiment analysis whereby 'neutral sentiment tweets' had no sentiments about war or its outcome, 'positive sentiment tweets' were evidenced by peace, world order, stopping war, and 'negative sentiment tweets' were evidenced by fear, anger, and opposition to the Russian invasion of Ukraine. Sentiment analysis showed that there were more neutral tweets compared to positive and negative sentiment tweets during March 2022. This work extends and combines existing techniques to assess how chatbots are influencing people in the online conversations about global issues (for example, Russia/Ukraine invasion). Our research has opened up avenues for future researchers to

understand chatbots effectively and how their development and deployment can help in ensuring cybersecurity.

Organizations today are incorporating better AI and AI-based chatbots for new avenues of sustainability, innovation, cost savings, business and revenue growth, and the overall sustainable development of businesses. Sustainable businesses have a positive impact on the economy, community, society, and environment. AI-based chatbots are tools for the sustainable development of businesses, since they increase efficiency, automate processes, and provide sustainable solutions for environmental, economic, and social issues: three pillars of sustainability [43]. These environmental, economic, and social factors should be tested by designers and developers for creating engaging uses of chatbots that can be utilized by global businesses sustainably. Developers should create reliable, sustainable, and practical chatbots by providing the interactive functions needed to develop humanized and natural conversations with these AI-based agents. These conversational agents should be based on natural language processing and machine-learning capabilities for creating a sustainable global impact on businesses, resulting in the best user experience and sustainable business development [44]. For example, Facebook messenger is powered by a computer program over AI, and this FB messenger chatbot tracks social media analytics and website traffic, thus boosting consumer confidence and consumer loyalty with brands and businesses. Another example is the homeowners and renters' insurance provider Lemonade using a customer onboarding chatbot, Maya, that can onboard customers in 90 s, as compared to online traditional insurers who take 10 min. In addition to Maya, Lemonade's claims chatbot Jim settles insurance claims within seconds, while traditional insurers may take anywhere between 48 h and 12 months to settle home insurance claims. Similarly, Marriott International's chatbot, ChatBotlr, is available through Facebook Messenger and Slack, and it helps Marriott Rewards members book their travel (plan for upcoming trips with suggestions linked from Marriot's digital magazine Marriott Traveler) to more than 4700 hotels.

Technological advancements may result in positive or negative impacts on sustainability. However, we argue that AI and AI-related chatbots help to make a positive impact. Through our research on social media AI-based chatbots and cybersecurity, we strive to benefit consumers through enhanced privacy and convenience, along with greater security, which positively impacts the sustainable development of businesses worldwide. Incorporating sustainable business practices into the development process of AI and AI-related technologies (e.g., chatbots on social media) will help to ensure the alignment of technologies with sustainable development principles. Utilizing the theory of planned behavior, current and future social media chatbots and cybersecurity researchers may wish to focus on the following research questions.

- Which AI-related technological advancements are better suited to promote environmental, economic, and social sustainability for global organizations?
- To what extent do privacy and security risks affect sustainability? How can social media chatbots prevent these risks and challenges?
- To what extent do technical factors (e.g., infrastructure, design) affect environmental, economic, and social sustainability for global organizations?
- Which sustainable business models can be developed and evaluated for AI and AI-related technology adoption, including circular and sharing economy models?
- What are the drivers and barriers to promoting sustainable AI-based systems, and how can AI-technology adoption support sustainable business practices globally?

## 5. Conclusions and Future Directions

Social media has made it possible for people around the world to communicate with each other freely and has reduced time and space constraints. At the same time, it has proved to be a useful tool to detect threats, both national and organizational, and subvert them in a timely manner. Future work entails automating the process of retrieving tweets from Twitter space and automating the sentiment analysis process. Expanding the work to

other social media sites, such as Reddit, etc., will help to expand the scope of the project. In a global world, threats can emanate from any part of the world and in any language. Future work needs to be conducted in terms of language modeling in languages other than English with a specific focus on Russian, Chinese, and Arabic. The chatbot developed on the Bot Libre platform needs to be refined in order to converse more naturally on social media. It needs to be more accurate in starting chat conversations with potentially threatening individuals and organizations in order to extract more information from these potential malicious sources. We expect future researchers to come up with innovative ideas and methods to fill the gaps in the current knowledge domain.

**Author Contributions:** A.A. (Amit Arora), A.A. (Anshu Arora), J.M.; methodology, A.A. (Amit Arora); software, A.A. (Amit Arora); validation, A.A. (Anshu Arora), A.A. (Amit Arora), J.M.; formal analysis, A.A. (Amit Arora); investigation, A.A. (Amit Arora), A.A. (Anshu Arora), J.M.; resources, A.A. (Amit Arora); data curation, A.A. (Amit Arora), A.A. (Anshu Arora).; writing—original draft preparation, A.A. (Amit Arora), A.A. (Anshu Arora), J.M.; writing—review and editing, A.A. (Anshu Arora), A.A. (Amit Arora), J.M.; visualization, A.A. (Anshu Arora).; supervision, J.M.; project administration, J.M.; funding acquisition, A.A. (Amit Arora). All authors have read and agreed to the published version of the manuscript.

**Funding:** This research was supported by the Office of the Undersecretary of Defense for Research and Engineering (OUSD (R&E)), United States Department of Defense (DoD) under the following agreement: HQ003421F0013, University of Maryland, College Park, "Pilot Projects for the ARLIS Intelligence & Security University Research Enterprise (INSURE) Academic Consortia".

**Conflicts of Interest:** The authors declare no conflict of interest.

## Appendix A

**The code for Twitter API credentials and the extraction of tweets is below.**

```
# Twitter API credentials
consumerKey = log ['key'] [0]
consumerSecret = log ['key'] [1]
accessToken = log ['key'] [2]
accessTokenSecret = log ['key'] [3]
bearer_token = log ['key'] [4]

accessToken

# Create the authentication object
authenticate = tweepy.OAuthHandler (consumerKey, consumerSecret)
# Set the access token and access token secret
authenticate.set_access_token (accessToken, accessTokenSecret)
# Create the API object while passing in the auth information
api = tweepy.API (authenticate, wait_on_rate_limit = True)

# Extract 2000 tweets
posts = [status for status in tweepy.Cursor (api.search, q = 'russia', tweet_mode = 'extended',
    lang = 'uk', retweeted = False, truncated = False).items(20)]
```

**The code for noise removal is below.**

```
#Create a function to clean the tweets
def cleanTxt (text):
text = re.sub (r'@[A-Za-z0-9]+', '', text) # Removed @mentions
text = re.sub (r'#', '', text) #Removing the "#" symbol
text = re.sub (r'RT[\s]+', '', text) # Removing RT
text = re.sub (r'https?:\/\/\S+', '', text) # Remove the hyper link
```

```
text = re.sub (r':[\s]+', ", text) # Removing:
text = text.lstrip ()

return text

#Cleaning the text
df ['Tweets'] = df ['Tweets'].apply (cleanTxt)
#Show the cleaned text
df
```

**The algorithm for finding tweets with a negative orientation is below.**

1.    Procedure **findOrientation**
2.    (*frequent_markers, compendium_of_samples*)
3.     begin
4.       **for** each marker $a_i$ in *frequent_markers*
5.        **for** each sample $b_j$ in *compendium_of_samples*
6.         if($a_i = b_j$ {
7.          if(**searchForNegative**($a_i$)! = false){
8.           $compendium\_of\_negative_{markers} = b_j$}
9.       **endfor**
10.      **endfor**

11.  **End**

1.    Procedure **searchForNagative**
2.    (*marker, sentiwordnet*)
3.     begin
4.       **for** each synset $s_i$ in *sentiwordnet*
5.       if(*marker* = $s_i$){
6.        if($s_{negative} > s_{positive}$)
7.         return *marker*}
8.          else {
9.        return false}
10.      **endfor**
11.   **end**

      **The codes to classify the subjectivity and polarity of tweets, form a word cloud, and form a scatterplot of subjectivity vs. polarity are below.**

```
# Create a function to obtain the subjectivity
def getSubjectivity (text):
return TextBlob (text).sentiment.subjectivity

# Create a function to obtain the polarity
def getPolarity (text):
return TextBlob (text).sentiment.polarity

# Create two new columns
df ['Subjectivity'] = df ['Tweets'].apply(getSubjectivity)
df ['Polarity'] = df ['Tweets'].apply(getPolarity)

# Show the new dataframe with the new columns
df

# Plot The Word Cloud
allwords = ' '.join ([twts for twts in df['Tweets']])
```

```
wordCloud = WordCloud (width = 1000, height = 600, random_state = 21,
max_font_size = 119).generate (allwords)
plt.imshow (wordCloud, interpolation = "bilinear")
plt.axis ('off')
plt.show ()

# Create a function to compute the negative, neutral and positive analysis
def getAnalysis (score):
if score < 0:
return 'Negative'
elif score == 0:
return 'Neutral'
else:
return 'Positive'

df ['Analysis'] = df [ 'Polarity' ].apply (getAnalysis)

# Show the dataframe
df

# Print all of the positive tweets
j = 1
sortedDF = df.sort_values (by = ['Polarity'])
for i in range (0, sortedDF.shape [0]):
if (sortedDF ['Analysis'][i] == 'Positive'):
print (str(j) + ') '+sortedDF['Tweets'][i])
print ()
j = j + 1

# Print all of the negative tweets
j = 1
sortedDF = df.sort_values (by = ['Polarity'], ascending = 'False')
for i in range (0, sortedDF.shape [0]):
if (sortedDF ['Analysis'] [i] == 'Negative'):
print (str(j) + ') '+sortedDF ['Tweets'] [i])
print ()
j = j + 1

# Plot the polarity and subjectivity
plt.figure (figsize = (8, 6))
for i in range (0, df. shape [0]):
plt.scatter (df ['Polarity'][i], df ['Subjectivity'] [i], color = 'Blue')
plt.title ('Sentiment Analysis')
plt.xlabel ('Polarity')
plt.ylabel ('Subjectivity')
plt.show ()

# Obtain the percentage of positive tweets
ptweets = df [df.Analysis == 'Positive']
ptweets = ptweets ['Tweets']

round ((ptweets.shape [0]/df.shape [0]) * 100, 1)

# Obtain the percentage of negative tweets
ntweets = df [df.Analysis == 'Negative']
ntweets = ntweets ['Tweets']
```

```
round ((ntweets.shape [0]/df.shape [0]) * 100, 1)

# Show the value counts

df ['Analysis'].value_counts ()

# Plot and visualize the counts
plt.title ('Sentiment Analysis')
plt.xlabel ('Sentiment')
plt.ylabel ('Counts')
df ['Analysis'].value_counts ().plot (kind = 'bar')
plt.show ()
```

**The code for developing the chatbot on the Bot Libre platform is below.**

```
<script type = 'text/javascript' src = 'https://www.botlibre.com/scripts/sdk.js'>
</script>
<script type = 'text/javascript' src = 'https://www.botlibre.com/scripts/game-
sdk.js'></script>
<script type = 'text/javascript'>
SDK.applicationId = "6191571217345391239";
SDK.backlinkURL = "http://www.botlibre.com/login?affiliate=allanmuir1";
var sdk = new SDKConnection ();
var user = new UserConfig ();
user.user = "allanmuir1";
user.token = "1393605116044980714";
sdk.connect (user, function () {
var web = new WebChatbotListener ();
web.connection = sdk;
web.instance = "41557310";
web.instanceName = "allantwitter bot";
web.prefix = "botplatform";
web.caption = "Chat Now";
web.boxLocation = "bottom-right";
web.color = "#009900";
web.background = "#fff";
web.css = "https://www.botlibre.com/css/chatlog.css";
web.gameSDKcss = "https://www.botlibre.com/css/game-sdk.css";
web.buttoncss = "https://www.botlibre.com/css/blue_round_button.css";
web.version = 8.5;
web.bubble = true;
web.backlink = true;
web.showMenubar = true;
web.showBoxmax = true;
web.showSendImage = true;
web.showChooseLanguage = true;
web.avatar = true;
web.chatLog = true;
web.popupURL = "https://www.botlibre.com/chat?&id=41557310&embedded=
true&chatLog=true&facebookLogin=false&application=6191571217345391239&
user=allanmuir1&token=1393605116044980714&bubble=true&menubar=true&
chooseLanguage=true&sendImage=true&background=%23fff&prompt=You+
say&send=Send&css=https://www.botlibre.com/css/chatlog.css";
web.createBox();
});
</script>
```

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
