# Peer review of "Developing Chatbots for Cyber Security: Assessing Threats through Sentiment Analysis on Social Media"

_sustainability, doi:10.3390/su151713178_

Round 1

Reviewer 1 Report

The following are my comments for improvement of this paper:

1.       The authors state – “[12] (pp. 373-383) states that “Intrapersonal chatbots exist…..” Consider updating the choice of words to Adamopoulou et al. [12] states that ……..

2.       In Section 3.1, the authors state – “Any keyword that can be deemed as a threat (e.g., hate speech, defense related, etc.) can be searched on Twitter platform using the chatbot” Please rewrite this part to describe the specific methodology and its step-by-step working that has been implemented in the chatbot to detect hate speech and defense-related Tweets.

3.       In Section 3.1.4, the authors state that the sentiment was detected on a scale of -1 to +1. Why was the VADER approach for sentiment analysis not used, which can detect sentiments on a scale of +4 to -4?

4.       The data preprocessing section should be rewritten to specifically highlight the steps that were followed. In addition to this, if stemming and/or lemmatization were performed, it should be described. If it was not performed, please justify why it was not applicable in this context

5.       The authors have discussed the importance of sentiment analysis on Twitter in the Introduction section. However, multiple papers cited in this context are not recent works in this field. Consider citing some recent works in sentiment analysis, such as https://doi.org/10.3390/bdcc7020116 and https://doi.org/10.1016/j.technovation.2022.102666

6.       A comparison with prior works is missing: Please include a comparative study (qualitative and quantitative) with prior works in this field to highlight the novelty of this work

Author Response

RESPONSE TO REVIEWER #1

Revised Version of Manuscript Sustainability-2562976

Title: Developing Chatbots for Cyber Security: Assessing Threats through Sentiment Analysis on Social Media

Introduction

Thank you for offering constructive comments to improve our manuscript.  They helped us a great deal to enhance the quality and contribution of the manuscript in several important ways. In this revision of our manuscript, we have tried to address your concerns to the best of our abilities. The following gives details regarding how we have addressed each of your comments.  We reproduce your comments in italics and provide our response below (in regular text) for each comment.

  1. The authors state – “[12] (pp. 373-383) states that “Intrapersonal chatbots exist…..” Consider updating the choice of words to Adamopoulou et al. [12] states that ……..

Thank you very much for pointing this out. As suggested, we have corrected this in the manuscript.

  1. In Section 3.1, the authors state – “Any keyword that can be deemed as a threat (e.g., hate speech, defense related, etc.) can be searched on Twitter platform using the chatbot” Please rewrite this part to describe the specific methodology and its step-by-step working that has been implemented in the chatbot to detect hate speech and defense-related Tweets.

Thank you for the comment. We have added a paragraph on Page 5 of the manuscript mentioning at least three different sources of military and defense-related terms.

  1. In Section 3.1.4, the authors state that the sentiment was detected on a scale of -1 to +1. Why was the VADER approach for sentiment analysis not used, which can detect sentiments on a scale of +4 to -4?

We have added a paragraph on Page 8 of the manuscript which briefly describes the two lexicon-based approaches, TextBlob and VADER. Since TextBlob calculates both polarity and subjectivity of the text, we have justified the use of this approach in our manuscript. We hope that you find this explanation acceptable. 

  1. The data preprocessing section should be rewritten to specifically highlight the steps that were followed. In addition to this, if stemming and/or lemmatization were performed, it should be described. If it was not performed, please justify why it was not applicable in this context

Section 3.1.2. Data Pre-Processing on Page 6 of the manuscript has been re-arranged to highlight the steps as (a), (b), (c). Part (b) has been re-written to properly explain the pre-processing to corpus of tweets. Additionally, part (c) has been added to include and highlight the lemmatization process in TextBlob.

  1. The authors have discussed the importance of sentiment analysis on Twitter in the Introduction section. However, multiple papers cited in this context are not recent works in this field. Consider citing some recent works in sentiment analysis, such as https://doi.org/10.3390/bdcc7020116 and https://doi.org/10.1016/j.technovation.2022.102666

Thank you very much for referring us to these articles. These citations have been added in the Introduction section on Page 2 of the manuscript. 

  1. A comparison with prior works is missing: Please include a comparative study (qualitative and quantitative) with prior works in this field to highlight the novelty of this work

Thank you for this suggestion. This helped us to strengthen the justification and contribution of our paper. We have added a new paragraph (pages 4 and 5) at the end of “Background and Related Work” section to highlight the novelty and contributions of our paper. Additionally, we have added specific research questions and the theoretical basis of our paper on page 2 of the manuscript.

We hope these additions help in enhancing the quality of our paper.

Conclusion
Thank you very much for your constructive comments and we appreciate the opportunity to submit a revised version of the manuscript. We have gone through the contents of the revised paper to ensure that the language and any spell checks are taken care of. We hope you find this version meeting the requirements for publication.

Reviewer 2 Report

The context of the research is clearly described and is appropriate to the research topic. The references and citations used in the theoretical framework are adequate, current and offer relevant information for the reader. Previous works that support the development of the research are detailed and each of the contents and variables of analysis in the research are adequately defined.

The objectives of the research are defined. However, the research questions that arise with the development of this work are not included, nor are they linked to the objectives. It is recommended to improve this aspect and include the research questions and the theoretical bases on which the construction of the objectives is based.

Referring to the methodological heading, there are some key aspects necessary to include for a better understanding of the document. It is recommended to include some mention of the paradigm on which this work is based, together with the description of the research design used, since there is only a description of the development and implementation of the chatbot and the procedures for carrying out the data analysis.

Regarding the results shown in figures 3, 4 and 5, they are only described as examples, but it would be necessary for a better understanding of the work, to detail and conclude what results have been obtained with these images. It should detail what frequencies of words, which are the most prominent and why, within those shown in figure 3. In figure 4, it is suggested to give examples with relevant tweets that show the aforementioned result of "represents specific tweets with a lot of subjectivity and with a negative polarity”. Similarly, Figure 5 should give concrete examples of tweets that reinforce the greater concurrence of neutral content, as opposed to negative and positive ones.

Finally, a deep improvement of the general discussion section is recommended, since only generic ideas on the use of chatbots and artificial intelligence are mentioned, including their potentialities, which have not been studied in this work, without including any reference to the results and experience shown in this work. It is necessary to highlight the findings found in the investigation, offering a link with the proposed research objectives, since there is no clear connection between both aspects within the discussion and conclusions.

Author Response

RESPONSE TO REVIEWER #2

Revised Version of Manuscript Sustainability-2562976

Title: Developing Chatbots for Cyber Security: Assessing Threats through Sentiment Analysis on Social Media

Introduction

Thank you for offering constructive comments to improve our manuscript.  They helped us a great deal to enhance the quality and contribution of the manuscript in several important ways. In this revision of our manuscript, we have tried to address your concerns to the best of our abilities. The following gives details regarding how we have addressed each of your comments.  We reproduce your comments in italics and provide our response below (in regular text) for each comment.

  1. The context of the research is clearly described and is appropriate to the research topic. The references and citations used in the theoretical framework are adequate, current and offer relevant information for the reader. Previous works that support the development of the research are detailed and each of the contents and variables of analysis in the research are adequately defined.

We greatly appreciate your favorable assessment.

  1. The objectives of the research are defined. However, the research questions that arise with the development of this work are not included, nor are they linked to the objectives. It is recommended to improve this aspect and include the research questions and the theoretical bases on which the construction of the objectives is based.

Thank you for the comment. As suggested, we have modified the section “Introduction” [Page 2] where research questions have been added and linked to the objectives. Theory of Planned Behavior is used as a theoretical basis of the research questions / objectives (Safa et al. 2015; 2016; 2019; Siyongwana, 2022). General Discussion section [Pages 10 - 11] is modified in relation to the theoretical framework of the research.

We hope you find the revisions acceptable.

  1. Referring to the methodological heading, there are some key aspects necessary to include for a better understanding of the document. It is recommended to include some mention of the paradigm on which this work is based, together with the description of the research design used, since there is only a description of the development and implementation of the chatbot and the procedures for carrying out the data analysis. 

We have revised the “Methods and Data Analysis” section of the paper to include Design of Experiments as the research design methodology used in our paper. Please refer to pages 4-5. 

  1. Regarding the results shown in figures 3, 4 and 5, they are only described as examples, but it would be necessary for a better understanding of the work, to detail and conclude what results have been obtained with these images. It should detail what frequencies of words, which are the most prominent and why, within those shown in figure 3. In figure 4, it is suggested to give examples with relevant tweets that show the aforementioned result of "represents specific tweets with a lot of subjectivity and with a negative polarity”. Similarly, Figure 5 should give concrete examples of tweets that reinforce the greater concurrence of neutral content, as opposed to negative and positive ones.

Thank you very much for your suggestions. Figures 3, 4, and 5 have been explained in detail along with examples of neutral, positive, and negative tweets during March 2022 highlighting global tensions during the Russian invasion of Ukraine. Please refer to Pages 8, 9, and 10 along with Table 1. 

  1. Finally, a deep improvement of the general discussion section is recommended, since only generic ideas on the use of chatbots and artificial intelligence are mentioned, including their potentialities, which have not been studied in this work, without including any reference to the results and experience shown in this work. It is necessary to highlight the findings found in the investigation, offering a link with the proposed research objectives, since there is no clear connection between both aspects within the discussion and conclusions. 

Thank you for your comments and suggestions. We have strengthened the “General Discussion” section [Pages 10-12] in order to explain the results of our work in light of research questions, objectives, and theoretical framework.

Conclusion

Thank you very much for your constructive comments and we appreciate the opportunity to submit a revised version of the manuscript. We have gone through the contents of the revised  paper to ensure that the language and any spell checks are taken care of. We hope you find this version meeting the requirements for publication.

Reviewer 3 Report

In the manuscript entitled "Developing Chatbots for Cyber Security: Assessing Threats 2 through Sentiment Analysis on Social Media", authors  presented  an  AI based   chatbot  to communicate, and deploy it on social media sites  for cyber security. They used sentiment analysis strategy to  analyze  Twitter datasets to predict cyber-attacks and future threats. The research is clearly presented with good dicussion on results.   This reviewer find the manuscript very intersting and  believe that it can accepted for  publication. One suggestion is that the quality of the illustrations and plots should be improved. 

Author Response

RESPONSE TO REVIEWER #3

Revised Version of Manuscript Sustainability-2562976

Title: Developing Chatbots for Cyber Security: Assessing Threats through Sentiment Analysis on Social Media

Introduction

Thank you for offering constructive comments to improve our manuscript.  They helped us a great deal to enhance the quality and contribution of the manuscript in several important ways. In this revision of our manuscript, we have tried to address your concerns to the best of our abilities. The following gives details regarding how we have addressed each of your comments.  We reproduce your comments in italics and provide our response below (in regular text) for each comment.

  1. In the manuscript entitled "Developing Chatbots for Cyber Security: Assessing Threats 2 through Sentiment Analysis on Social Media", authors  presented  an  AI based   chatbot  to communicate, and deploy it on social media sites  for cyber security. They used sentiment analysis strategy to  analyze  Twitter datasets to predict cyber-attacks and future threats. The research is clearly presented with good dicussion on results.   This reviewer find the manuscript very intersting and  believe that it can accepted for  publication. One suggestion is that the quality of the illustrations and plots should be improved. 

We greatly appreciate your favorable assessment.

Conclusion

Thank you very much for your constructive comments and we appreciate the opportunity to submit a revised version of the manuscript. We hope you find this version meeting the requirements for publication.

Round 2

Reviewer 1 Report

The authors have revised their paper as per all my comments and feedback. I do not have any additional comments at this point. I recommend the publication of the paper in its current form. 

Reviewer 2 Report

Headings that had more significant deficiencies have been revised and refined. Important aspects of the design have been improved, such as the selection of the participants and the procedures used in the classification and analysis of the results. Effective methods and discussion opportunities related to the results of the project have been established. In my opinion, the work exhibits an appropriate level of quality and the discoveries made contribute significantly to the field of study.